# Contemporary Systemic Therapy Intensification for Prostate Cancer: A Review for General Practitioners in Oncology

**Anupam Batra** *\*[ID], **Daniel Glick and Mario Valdes**

Department of Oncology, Grand River Regional Cancer Centre, 835 King St. W., Kitchener, ON N2G 1G3, Canada; daniel.glick@grhosp.on.ca (D.G.); mario.valdes@grhosp.on.ca (M.V.)
\* Correspondence: anupam.batra@grhosp.on.ca

**Abstract:** Prostate cancer accounts for a significant proportion of cancer diagnoses in Canadian men. Over the past decade, the therapeutic landscape for the management of metastatic prostate cancer has undergone rapid changes. Novel strategies use hormonal agents, chemotherapy, homologous recombination repair inhibitors, and radioligand therapy or combination strategies in addition to androgen deprivation therapy. In this review, we summarize the available data addressing key therapeutic areas along the disease continuum and focus on practical aspects for general practitioners in oncology managing patients with metastatic prostate cancer.

**Keywords:** prostate cancer; ARAT; androgen receptor axis-targeted drugs; docetaxel; BRCA1/2; breast cancer gene 1/2; PARP inhibitors; poly ADP ribose polymerase inhibitors; prostate-specific membrane antigen; PSMA; CAGPO; Canadian Association of General Practitioners in Oncology

## 1. Introduction

Prostate cancer is one of the most frequently diagnosed neoplasms in men. In 2022, nearly 24,600 Canadians were diagnosed with prostate cancer, representing 20% of all new cancer cases in men. Moreover, 4600 Canadian men died from prostate cancer, representing 10% of all cancer deaths in males. The death rate for prostate cancer has been declining since 1994, reflecting improved treatment [1]. Approximately one-third of all prostate cancers are metastatic at the time of diagnosis (synchronous) or recur (metachronous) following definitive treatment. The main therapeutic strategy for metastatic prostate cancer involves androgen deprivation therapy (ADT) [1,2]; however, most patients will develop lethal castration resistance [3]. In the past decade, new androgen receptor axis-targeted (ARAT) drugs, namely abiraterone acetate plus prednisone (henceforth abiraterone), enzalutamide, apalutamide and darolutamide have been assessed across different stages of prostate cancer [4–11]. Their application early in the disease trajectory has resulted in both improved survival and maintenance of quality of life (QoL) in men with non-metastatic and metastatic prostate cancer among all age groups [12–16]. Recently, frontline combination therapy in metastatic castrate-resistant prostate cancer targeting both the androgen pathway and poly ADP ribose polymerase has improved outcomes for disease-harboring deleterious mutations in breast cancer gene −1 and −2 (BRCA1/2) along with ataxia telangiectasis (ATM). We also summarize existing and promising new strategies using radiopharmaceuticals such as radium-223 and prostate-specific membrane antigen (PSMA)-based therapy [17,18]. Herein, we review the literature addressing the rapidly changing paradigm for the management of metastatic prostate cancer by highlighting relevant studies of particular impact on clinical practice with a special focus on general practitioners in oncology who may prescribe and supervise such agents within the Canadian cancer system.

## 2. Metastatic Castrate-Sensitive Prostate Cancer

The benefits of surgical or pharmacologic castration with androgen deprivation therapy (ADT) were established in 1941 [12]. Metastatic prostate cancer was consequently considered to be either castration-sensitive or -resistant, referring to an early state of the disease susceptible to castration and a later state that is no longer vulnerable to testosterone suppression, respectively.

A paradigm shift occurred in 2015 with the advent of combination therapy, so-called "intensification beyond androgen deprivation therapy". Improved overall survival (OS) was demonstrated in studies assessing the efficacy of adding docetaxel or androgen biosynthesis inhibition with abiraterone acetate and prednisone along with blockade of the androgen receptor with enzalutamide or apalutamide. These data will be summarized below.

*A. Cytotoxic Drug Therapy*

The addition of docetaxel was the first agent to demonstrate favorable efficacy in this setting. When compared to ADT alone, docetaxel administered over six cycles every 21-days initiated within the first 120 days of ADT in patients with de novo metastatic disease was associated with improved median OS in patients with high-volume disease (median OS 44 months with combination vs 32.2 months with ADT alone). High-volume disease was defined as the presence of visceral metastasis or at least four bone metastases of which at least one had to be present outside of the spine and pelvis [13]. All other disease states with metastases that do not meet these criteria are referred to as low-volume. In 2016, another trial confirmed these results, with an improvement in median OS of 10 months [14] (Figures 1 and 2).

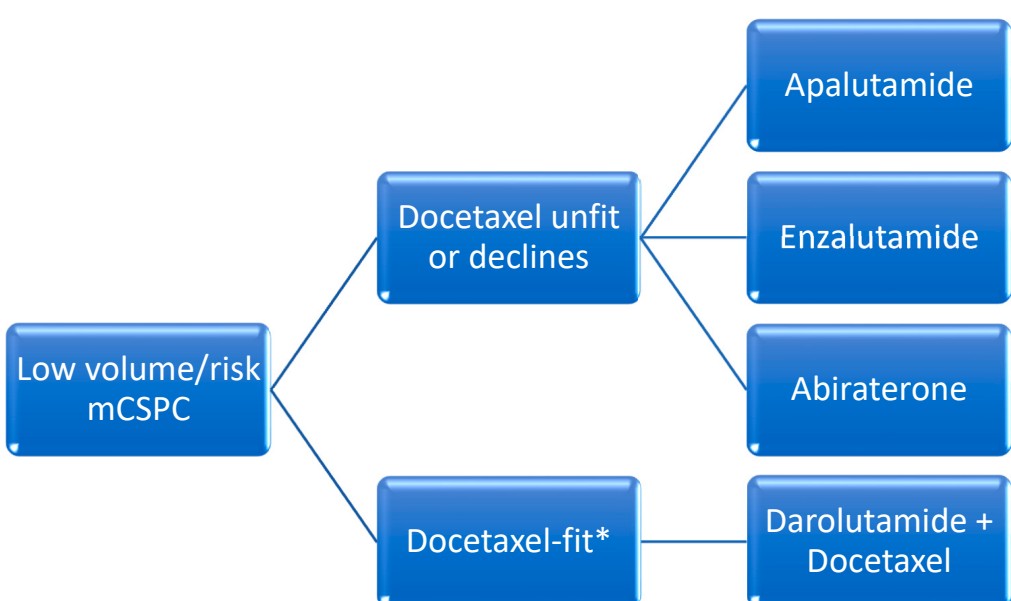

**Figure 1.** Proposed overview for the management of low-volume or -risk metastatic castration-sensitive prostate cancer (CSPC). High risk definition (from LATITUDE) [7]: at least 2 of the following: ≥3 bone metastases, visceral metastases, Gleason score ≥8. High volume definition (from CHAARTED) [13]: at least 1 of the following: ≥4 bone lesions with ≥1 beyond the vertebral bodies and pelvis or visceral metastases. * = Data for the use of docetaxel in low-volume disease is limited, see text for further discussion.

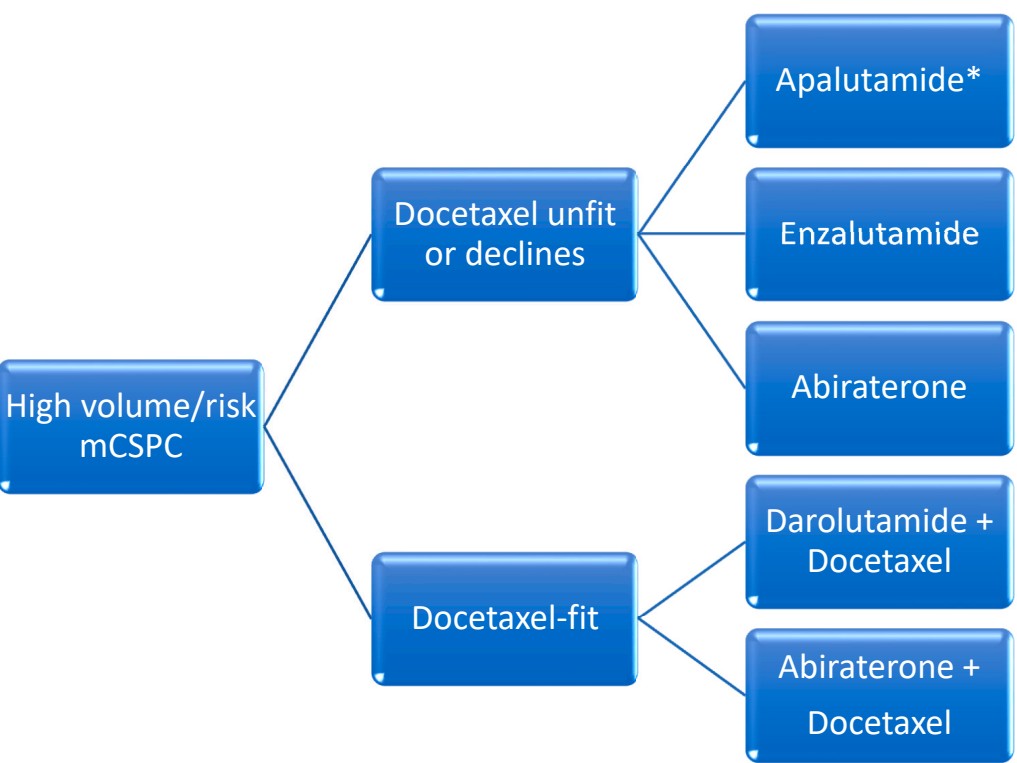

**Figure 2.** Proposed overview for the management of high-volume or risk metastatic castration-sensitive prostate cancer (CSPC). High risk definition (from LATITUDE) [7]: at least 2 of the following: ≥3 bone metastases, visceral metastases, Gleason score ≥8. High volume definition (from CHAARTED) [13]: at least 1 of the following: ≥4 bone lesions with ≥1 beyond the vertebral bodies and pelvis or visceral metastases. * = Rezvilutamide where available.

*B. Oral Hormonal Agents and ADT ("Doublet-Therapy")*

Abiraterone acetate demonstrated an OS benefit when administered until disease progression in patients with de novo metastatic disease and high-risk features defined as two of the following: Gleason score 8 or higher, visceral disease and at least three bone metastases. Radiographic progression free survival (rPFS) was 33 months in the abiraterone group versus 14.8 months with placebo. The reduction in the risk of radiologic progression was 53%. Median OS was 53.3 months in patients receiving abiraterone versus 36.5 months in patients receiving placebo, with a relative improvement in OS of 33% [7].

Enzalutamide was associated with improved survival in patients with de novo metastatic disease in the ARCHES trial [16]. The primary endpoint, median rPFS, was not reached with enzalutamide plus ADT versus 19 months with placebo plus ADT, with a significant reduction in risk of 61%. The risk reduction for death in the final report was significantly decreased by 33%. In the ENZAMET trial, patients randomized to ADT and enzalutamide versus ADT and non-steroidal antiandrogens demonstrated a survival advantage that favored enzalutamide [19,20]. Median OS was not reached after a median follow-up period of 44 months in ARCHES and 68 months in ENZAMET. A subgroup analysis of ENZAMET patients who had received prior docetaxel chemotherapy did not demonstrate any incremental OS benefit.

Apalutamide was assessed in the TITAN trial, which randomized patients with mCSPC to ADT plus apalutamide versus placebo plus ADT and primary endpoint of rPFS. In this study, about 10% of patients had received prior docetaxel chemotherapy. Furthermore, about one-third of patients did not have high-volume disease using the CHAARTED definition. Investigators showed that rPFS was improved by 52% in the apalutamide group whereas OS was improved by 35% and median OS was not reached for the apalutamide group versus 52.2 months in patients receiving ADT alone [21,22].

A novel second-generation drug unavailable in North America is rezvilutamide. This compound's efficacy and safety were assessed in the phase III CHART trial, which enrolled men with high-volume mCSPC who had never received chemotherapy and randomized them to ADT plus rezvilutamide versus ADT plus bicalutamide. Notably, 90% of patients were Chinese. Improved rPFS was demonstrated with rezvilutamide compared to bicalutamide (median rPFS not reached) versus 25.1 months (HR 0.44 [95% CI 0.33–0.58]; $p < 0.0001$). Rezvilutamide significantly improved overall survival compared to bicalutamide (HR 0.58 (95% CI 0.44–0.77); $p = 0.0001$), whereas median OS was not reached for either arm at the time of publication. The most common grade $\geq 3$ adverse events were hypertension (8% versus 7%), hypertriglyceridemia (7% versus 2%), weight gain (6% versus 4%), anemia (4% versus 5%) and hypokalemia (3% versus 1%), with rezvilutamide compared to bicalutamide, respectively [23].

## C. Docetaxel with Oral Hormonal Agents and ADT ("Triplet-Therapy")

As of 2024, none of these strategies have been formally compared to one another in a clinical trial and are considered alternatives. The most recent development in this setting is the combination of ADT, docetaxel chemotherapy and either abiraterone or darolutamide.

### i. Darolutamide with Docetaxel and ADT

ARASENS was a phase III trial that assessed the efficacy and safety of darolutamide in patients with mCSPC who were candidates for ADT and docetaxel. Participants were randomized to ADT and six cycles of docetaxel and darolutamide versus ADT, docetaxel and placebo. The primary endpoint was OS and was not reached for the darolutamide group versus 48.9 months for placebo, with a significant improvement in the risk of death of 32%. The study did not stratify patients by risk or volume of disease [24]. A subsequent report assessed therapeutic benefits in all volume and risk settings, except for low-volume disease [25].

### ii. Abiraterone with Docetaxel and ADT

Abiraterone was assessed in the PEACE-1 trial which compared ADT, docetaxel and abiraterone vs ADT and docetaxel in patients with de novo metastatic hormone sensitive prostate cancer. Patients were stratified by disease volume as per the CHAARTED criteria mentioned above. Participants with high-volume disease had significant improvement in OS (median 61 vs. 42 months), whereas patients with low-volume did not [26]. Achievement of undetectable 8-month PSA after initiation of ADT with abiraterone and docetaxel predicted improvement in rPFS and OS [27].

### iii. Triplet Therapy Data Limitations

Given the lack of available head-to-head comparisons between triplet (darolutamide or abiraterone with docetaxel and ADT) and doublet strategies (ARAT and ADT), the use of either approach is currently accepted as a reasonable evidence-based alternative. In all cases, a careful discussion of the available evidence is recommended to enable patients to make well-informed decisions. A systematic review of all the main phase III trials indirectly compared these different approaches and concluded that the best OS and radiologic PFS occurred with abiraterone-based triplet therapy vs. ADT alone, notwithstanding study limitations. However, this is still an indirect comparison of clinical trials with different methodologies and disease characteristics such as high and low volume and/or risk, and the information must therefore be interpreted with caution [25].

## D. Radiotherapy for Low-Volume Disease

The role of radiotherapy in newly diagnosed metastatic prostate cancer was investigated in the STAMPEDE trial, which compared systemic therapy alone to systemic therapy plus radiotherapy to the prostate using 55 Gy in 20 fractions or 36 Gy in 6 fractions. Systemic therapy consisted of lifelong ADT with or without docetaxel (18%). Metastatic burden was

defined per the CHAARTED criteria above. In patients with low-burden disease, but not high-burden, radiotherapy improved 5-year overall survival (HR 0.64, 95% CI 0.52–0.79; *p* < 0.001), failure-free survival and progression free survival [28,29]. These data suggest that prostate radiotherapy should be recommended as a standard of care for these men.

### 3. Non-Metastatic Castrate-Resistant Prostate Cancer

High-risk non-metastatic castrate-resistant prostate cancer is a newly classified disease state which meets the following definition:

1.  Histologically or cytologically confirmed adenocarcinoma of the prostate;
2.  Increasing PSA despite castrate levels of testosterone;
3.  PSA doubling time of 10 months or less;
4.  No previous or current evidence of metastatic disease as assessed by computed tomography or magnetic resonance imaging of the chest abdomen and pelvis and with whole-body radionuclide bone scan.

Targeting the androgen receptor with second-generation inhibitors such as enzalutamide, apalutamide or darolutamide has demonstrated improvement in progression-free and OS in patients with high-risk nmCRPC (Figure 3).

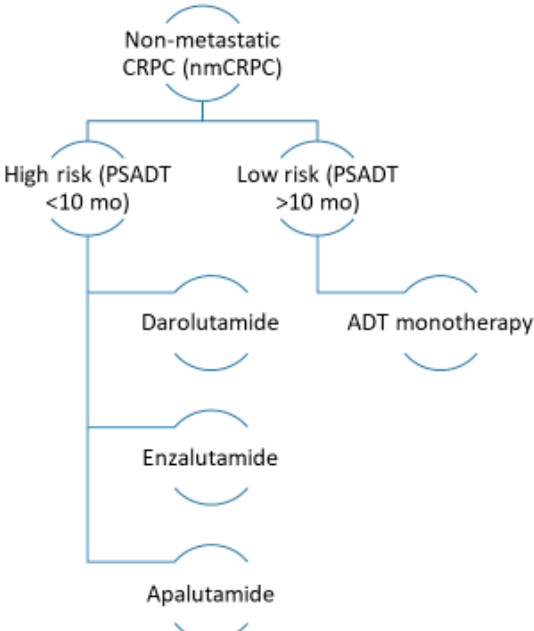

**Figure 3.** Proposed overview for the management of non-metastatic castration-resistant prostate cancer (CRPC). Mo: months; PSADT: prostate-specific antigen doubling time.

In the SPARTAN trial, apalutamide and ADT were assessed against placebo and ADT and had a median follow-up of 52 months and 428 deaths on study. Treatment duration with apalutamide was 32.9 months versus 11.5 for placebo. Furthermore, apalutamide improved OS to 73.9 months compared to 59.9 months for placebo. Apalutamide was started at a dose of 240 mg by mouth daily [30,31].

In the ARAMIS trial, darolutamide was associated with a median metastasis-free survival (primary endpoint) of 40.4 months versus 18.4 months with placebo. Median OS was not reached for either group in the first analysis. The updated survival analysis reported at the 2020 annual meeting of the American Society of Clinical Oncology (ASCO) showed a 31% improvement in the risk of death. Darolutamide is started at a dose of 600 mg by mouth twice a day [32,33].

Enzalutamide was assessed in the PROSPER trial, which showed a median OS of 67.0 months in the enzalutamide group versus 56.3 months with placebo. The median

duration of treatment was 33.9 months with enzalutamide vs. 14.2 months with placebo. Enzalutamide's starting dose was 160 mg by mouth daily [34,35].

## 4. Metastatic Castrate-Resistant Prostate Cancer

Ultimately, the majority of men with metastatic prostate cancer will develop progressive disease despite adequate castration. Two trials have demonstrated a marginal survival benefit for patients remaining on LHRH agonists while receiving systemic therapy for metastatic castrate-resistant disease [36,37]. Furthermore, subsequent therapeutics used for castrate-resistant prostate cancer have been studied in men undergoing androgen suppression. In light of the absence of prospective data and a favorable benefit-to-risk profile for LHRH analogues, these agents should be continued in this population.

This section will address intensification strategies beyond the modulation of the luteinizing hormone (Figure 4). We will introduce several novel compounds, including poly ADP ribose polymerase inhibitors and radioligand-based therapeutics (currently used in later lines) given their expected incorporation into standard practice in select situations.

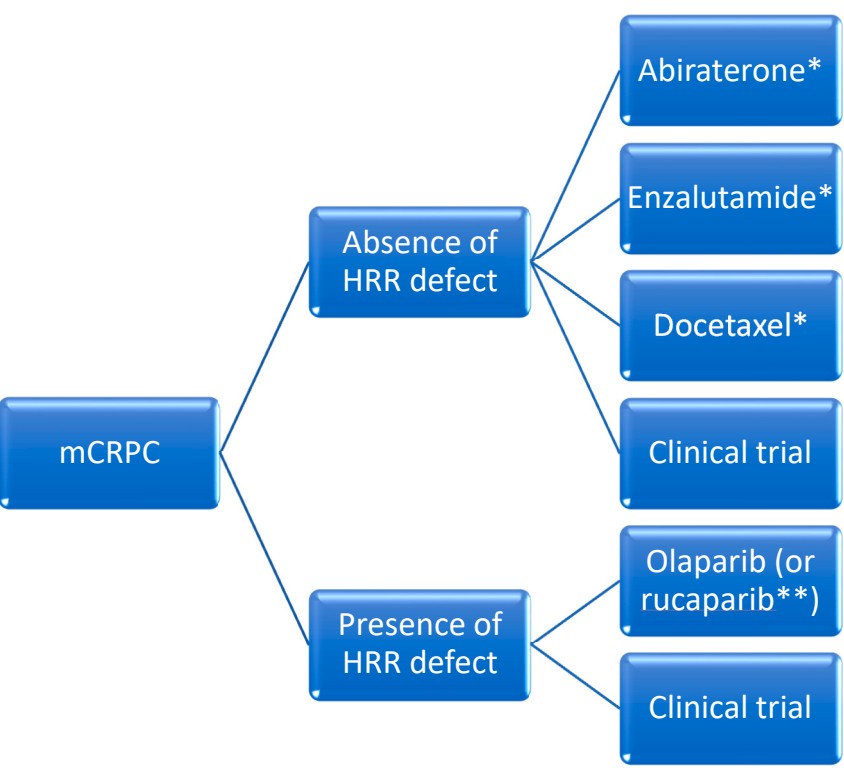

**Figure 4.** Proposed overview for the management of metastatic castration-resistant prostate cancer in the frontline setting. HRR: homologous recombination repair. * If not received previously; ** Notwithstanding access to rucaparib.

*A. Cytotoxic Drug Therapy*

### 4.0.1. Docetaxel

Standard chemotherapy for men with metastatic castrate-resistant disease is docetaxel at 75 mg/m$^2$ with prednisone 5 mg bid every three weeks for ten cycles. TAX 327 compared docetaxel (given either every three weeks or weekly) plus daily prednisone with mitoxantrone plus prednisone [38]. Men receiving docetaxel every three weeks had a 25% relative improvement in OS compared to mitoxantrone. Median survival was 16.5 months in the mitoxantrone group and 18.9 months in the group given docetaxel every 3 weeks. A PSA decrease of at least 50% was observed in 45% of docetaxel recipients. Pain reduction was observed in 35% of the docetaxel group compared to 22% in the mitoxantrone group ($p = 0.01$), whereas improvement in quality of life was found in 22% of docetaxel

recipients compared to 13% in the mitoxantrone group (*p* = 0.009). Adverse events were more common in recipients of docetaxel. Independent prognostic factors used to stratify response to docetaxel include visceral metastases, pain, anemia (Hb < 13 g/dL), bone scan progression and prior estramustine. Further stratification into three different risk groups is as follows: low (0–1), intermediate (2) and high (3–4). Associated median OS is 25.7, 18.7 and 12.8 months, respectively [39].

Special consideration should be given to older adults with multiple co-morbidities and reduced function [40]. The use of a validated tool to assess pre-treatment probability of toxicity to chemotherapy in older adults, such as the Cancer and Aging Research Group's Chemotherapy Toxicity Tool, may be beneficial in select patients [41,42]. In men who may be unable to tolerate standard docetaxel at 75 mg/m$^2$ every 3 weeks, an alternate schedule such as docetaxel 50 mg/m$^2$ every two weeks carries a lower risk of severe AEs and a prolonged time to treatment failure [43].

### 4.0.2. Cabazitaxel

Cabazitaxel has activity in docetaxel-refractory cancers, as demonstrated in the TROPIC trial [44]. In this study, men with mCRPC and progression within 12 months or during docetaxel were randomized to cabazitaxel plus prednisone versus mitoxantrone plus prednisone. Patients received a maximum of 10 cycles of cabazitaxel at 25 mg/m$^2$ or mitoxantrone at 12 mg/m$^2$ along with prednisone. The primary endpoint of overall survival was improved (HR 0.70, 95% CI 0.59–0.83), whereas the secondary endpoint of PFS was also prolonged (HR 0.74, 95% CI 0.64–0.86). Both hematologic and non-hematologic toxicities were more common in the cabazitaxel arm compared to standard of care (68.2% versus 47.3%, *p* < 0.0002 and 57.4% versus 39.8%, *p* < 0.0002). A later subset analysis suggested that the observed survival benefit was greater in patients having been exposed most heavily to docetaxel previously when compared to lesser exposure (HR 0.51 versus 0.96 for patients having received >900 mg/m$^2$ versus less than 225 mg/m$^2$). Additional follow-up resulted in a greater likelihood of surviving at least 2 years with cabazitaxel [45].

Cabazitaxel is the optimal treatment strategy for men progressing quickly on an ARAT agent (<12 months) and prior docetaxel as shown in the CARD trial [46]. The study assessed cabazitaxel after docetaxel and one prior line of ARAT therapy (either abiraterone or enzalutamide) for progressive disease within one year. Cabazitaxel doubled rPFS versus an alternate agent and improved relative survival by 36%. Lastly, rPFS remained superior regardless of sequence, i.e., docetaxel followed by an ARAT agent or vice versa. Similar grade ≥3 adverse events frequencies were observed in each study arm (56 versus 52%); however, asthenia was more common the cabazitaxel arm (4 versus 2.4%) along with diarrhea, peripheral neuropathy and febrile neutropenia (3.2 versus none for all categories). In a later analysis of quality of life by Fizazi et al. [47], 220 men with moderate-to-severe pain at randomization receiving third-line cabazitaxel versus androgenic axis interference had improved pain responses based on patient-reported outcomes (46 versus 19%, *p* < 0.0001).

### B. Androgen Receptor Axis-Targeted Agents
### 4.0.3. Abiraterone Acetate Prior to Docetaxel

Men with docetaxel-naïve castrate-resistant disease (and no visceral metastases) were randomized to abiraterone acetate or placebo, both combined with prednisone, in the phase III COU-AA-302 trial [9]. OS and rPFS were the co-primary endpoints. Final analysis at a median of 49.2 months showed that OS was significantly improved (34.7 vs. 30.3 months, relative risk reduction of 19%) [48]. Adverse events (AEs) related to mineralocorticoid excess and liver function abnormalities were more frequent with abiraterone, but mostly mild. In a subset analysis, Roviello et al. showed that abiraterone was equally effective in adults older than 75 years of age when compared to their younger counterparts [49]. Neuropsychological consequences were assessed by Harland et al.; abiraterone delayed

median time to emotional wellbeing deterioration by more than 6 months when compared to prednisone [50].

### 4.0.4. Abiraterone Acetate after Docetaxel

Men with docetaxel-refractory disease were randomized to abiraterone with prednisone versus placebo and prednisone at a 2:1 ratio in the final analysis of the large phase III trial COU-AA-301 [51]. The primary endpoint was OS, whereas secondary endpoints were rPFS, biochemical progression, time to PSA and clinical progression. At a median follow-up of 20.2 months, median survival in the abiraterone group was 15.8 months versus 11.2 months with placebo (HR: 0.78, $p < 0.0001$). Secondary endpoints favored the abiraterone arm over placebo. The frequency of grade 3–4 adverse events did not differ significantly between groups; however, mineralocorticoid-related toxicities were higher with abiraterone and were mainly low-grade fluid retention, edema and hypokalemia.

### 4.0.5. Enzalutamide Prior to Docetaxel

In the PREVAIL study, men with docetaxel-naïve castrate-resistant disease (including a small number of patients visceral metastases) were randomized to enzalutamide or placebo [8]. Results showed a relative improvement in the co-primary endpoints of rPFS by 80% and OS by 30%. PSA decrease of at least 50% was observed in 78% of participants. The most common clinically relevant AEs were fatigue and hypertension. Notably, enzalutamide carries a small (<1%) risk of seizures and should be avoided in patients with risk factors for seizures such as epileptic disorder, prior cerebrovascular infarct, concomitant use of seizure threshold-lowering drugs or brain metastases. In a subset analysis, enzalutamide was equally effective and well tolerated in men older than 75 years of age [52]. There was no difference in efficacy for men with liver metastases [53].

### 4.0.6. Enzalutamide after Docetaxel

The AFFIRM trial randomized 1199 men with docetaxel-refractory mCRPC to enzalutamide versus placebo at a 2:1 ratio [54]. Although not required by the study, glucocorticoids were received by 30% of participants. The primary endpoint was OS, whereas secondary endpoints included PSA response, quality of life, soft tissue response, time to PSA and clinical progression. At a median follow-up of 14.4 months, median survival with enzalutamide was 18.4 months versus 13.6 months in the placebo group. Enzalutamide had activity in patients with visceral metastases irrespective of age, baseline pain and type of progression. Secondary outcomes also favored the enzalutamide group. Adverse event profile was similar between groups but with a lower incidence of grade 3–4 adverse events in recipients of enzalutamide. Notably, there was a <1% chance of seizures with enzalutamide versus none with placebo. Finally, health-related quality of life in a separate report was significantly improved across all domains when compared to placebo based upon the Functional Assessment of Cancer Therapy-Prostate (FACT-P) scores [55].

*C. Novel Therapeutic Strategies Targeting Poly ADP-Ribose Polymerase*
### 4.0.7. Monotherapy using Poly ADP-Ribose Polymerase (PARP) Inhibition

The first study to show a benefit for genetic testing and precision medicine in mCRPC was the PROfound randomized phase III trial [56,57]. Herein, men with mCRPC harboring a qualifying gene alteration in HRR with prior ARAT exposure received the PARP inhibitor olaparib compared to an alternate ARAT agent. Most participants were heavily pre-treated with one to two cytotoxic drug therapies and up to two prior ARAT agents. The primary endpoint was radiographic PFS as assessed by blinded independent central review in the BRCA1/2 or ATM-mutated group (Cohort A) and was significantly improved for the olaparib group (HR: 0.49, 95% CI: 0.38–0.63). Moreover, the final OS results demonstrated a significant improvement among men whose tumors harbored BCRA ½ or ATM mutations (Cohort A) ($p = 0.0175$; HR 0.69, 95% CI: 0.50–0.97). Men with any other HRR alteration (Cohort B) did not derive any significant benefit from olaparib. Cross over to olaparib due

to progression on the enzalutamide/abiraterone arm was 66% (*n* = 86/131). The most common adverse events were anemia (30.1% versus 17.7%) and fatigue (26.2% versus 20.8%) for olaparib compared to an alternate ARAT agent. Notably, 4.3% of olaparib recipients experienced venous thromboembolic disease compared to 0.8% among enzalutamide/abiraterone. There were no reports of myelodysplastic syndrome or acute myeloid leukemia.

TRITON3 was a randomized, open-label phase III study assessing the efficacy of rucaparib versus docetaxel or second-generation ARAT agent in mCRPC harboring a deleterious defect in homologous recombination repair [58]. All patients had a history of disease progression after treatment with one previous second-generation ARAT agent. The primary endpoint of rPFS for the *BRCA* subgroup was improved by 50% with rucaparib compared with physician's choice (*p* < 0.0001). Median interim OS in the *BRCA* subgroup for rucaparib versus docetaxel was numerically improved by 6 months (24 vs 18.9 months, *p* = 0.16). Similarly, median interim OS in the *BRCA* subgroup for rucaparib versus second-generation ARAT was numerically improved by 2 months (24.3 vs. 22.1 months, *p* = 0.40). However, OS results were immature (54% in the *BRCA* group). Lastly, 60% of patients in the rucaparib arm reported at least one severe toxic event. The most frequently reported toxicities in all treatment groups were asthenia and fatigue.

4.0.8. Combination Therapy Using an ARAT Agent and PARP Inhibitor

i. Enzalutamide and Talazoparib

TALAPRO-2 was a phase III study comparing enzalutamide and talazoparib to enzalutamide and placebo in men with previously untreated Mcrpc [59]. Patients were stratified by prior abiraterone or docetaxel treatment for mCSPC and HRR gene alteration status. The primary endpoint was rPFS with secondary endpoint of OS. A significant improvement in rPFS was demonstrated in the talazoparib and enzalutamide group versus placebo and enzalutamide (not reached vs. 21.9 months, *p* < 0.001). Further, a 55% relative improvement in rPFS was observed in HRR-deficient (*p* < 0.001) and 35% relative improvement for non-HRR-deficient patients by tumor tissue testing for patients treated with talazoparib and enzalutamide versus placebo and enzalutamide. No significant difference in OS was observed at interim analysis. The most common toxicities with talazoparib and enzalutamide were anemia (65.8%), neutropenia (35.7%) and fatigue (33.7%).

ii. Abiraterone and Niraparib

MAGNITUDE was another phase III study assessed the efficacy of abiraterone and niraparib versus abiraterone and placebo in men with mCRPC in all-comers who have not received prior therapy for mCRPC [60]. A statistically significant 47% risk reduction was observed in the primary endpoint of rPFS (*p* = 0.001). At the second interim analysis (IA2), with median follow-up of 2 years in the *BRCA*-positive subgroup, median rPFS favored abiraterone and niraparib over placebo and abiraterone by 45%. The most common AEs in patients who received niraparib and abiraterone versus abiraterone and placebo were musculoskeletal pain (44% versus 42%), fatigue (43% versus 30%), constipation (34% versus 20%), hypertension (33% versus 27%) and nausea (33% vs. 21%). Niraparib combined with abiraterone have received both U.S. FDA approval and Health Canada authorization with conditions.

iii. Abiraterone and Olaparib

The phase III study PROpel was designed to evaluate the efficacy of abiraterone and olaparib versus abiraterone and placebo based on rPFS and OS as first line therapy in mCRPCC [61]. Median OS was not statistically significant for patients treated with abiraterone and olaparib versus abiraterone and placebo (42.1 versus 34.7; *p* = 0.0544). There was a trend towards benefit in OS with abiraterone and olaparib in the HRR-deficient subgroup (median OS: NR vs. 28.5 months; HR = 0.66; 95% CI: 0.45–0.95). In the non-HRR-deficient subgroup, the median OS for abiraterone and olaparib was 42.1 months versus

38.9 months for abiraterone and placebo (not statistically significant). The most common AEs reported for olaparib and abiraterone were anemia, fatigue and nausea.

*D. Radiopharmaceutical Agents*

4.0.9. Radium-223

The ALSYMPCA trial assessed the efficacy of radium-223, a bone-targeting alpha-emitting radiopharmaceutical agent, in 921 men with symptomatic mCRPC who previously received or were unfit for docetaxel [62]. The primary endpoint of median OS was improved by 3.6 months (HR: 0.70, $p < 0.001$) when compared to standard of care. Secondary endpoints of time to first skeletal-related event, pain scores and QoL were also improved. Relevant toxicities were generally mild and included hematologic toxicity and diarrhea without a significant difference relative to controls. Furthermore, radium-223 was effective and tolerable in men regardless of previous docetaxel exposure [63]. Notably, concomitant use of radium-223 and abiraterone acetate showed significant toxicities such as an increased fracture risk and mortality, especially in adults without concurrent use of bone-modifying agent [64].

4.0.10. PSMA-Based Therapy

The TheraP trial was a randomized phase II study that included patients for whom cabazitaxel was deemed to be the most appropriate next standard treatment after progressive disease with docetaxel. Participants were selected based on positive $^{68}$Ga-PSMA-11 and $^{18}$FDG PET-CT and were randomized to receive $^{177}$Lu-PSMA-617 intravenously every 6 weeks for up to six cycles compared with cabazitaxel at 20 mg/m$^2$ for up to ten cycles. The primary endpoint of at least a 50% or more reduction in PSA was observed in 66% versus 37% for $^{177}$Lu-PSMA-617 versus cabazitaxel recipients, respectively, by intention-to-treat (95% CI: 16–42; $p < 0.0001$) [18].

The open-label phase III trial VISION compared $^{177}$Lu-PSMA-617 with standard of care (excluding chemotherapy, immunotherapy, radium-223 or investigational agents) in heavily pre-treated mCRPC (at least one prior ARAT agent and one or two prior taxanes) and PSMA-avid disease based on PET-CT. Participants receiving $^{177}$Lu-PSMA-617 had prolonged imaging-based PFS and OS when compared to standard of care. Grade 3 or above toxicities were observed at 52.7% with $^{177}$Lu-PSMA-617 compared to 38.0% for controls, respectively, whereas QoL was not significantly different [17].

**5. Putting It All Together: Practical Recommendations for General Practitioners in Oncology**

Most men with metastatic prostate cancer are candidates for intensification above androgen deprivation. Important considerations when prescribing and supervising ARAT agents include understanding efficacy (summarized above), addressing relevant toxicities, managing drug–drug interactions, determining progressive disease and understanding unique considerations for older adults (Figure 5). In this section, we plan to introduce the latter four concepts. In all cases, the standard recommendation is to monitor drug therapy on a continuous basis.

*A. Addressing Relevant Toxicities*

Abiraterone must be used with caution in patients with hypertension, underlying cardiac arrhythmia or ischemic heart disease, requiring multiple agents for optimal control. In the COU-AA-302 trial [9], mineralocorticoid-related toxic effects were more common in the abiraterone–prednisone group than in the prednisone-alone group, including cardiac disorder (19 versus 16%), hypertension (22% versus 13%), hypokalemia (17% versus 13%) and fluid retention (28% versus 24%) and were mostly mild grade. The authors recommend baseline ECG prior to initiation of abiraterone in all recipients along with optimization of cardiovascular risk factors by family physicians and/or cardiologists.

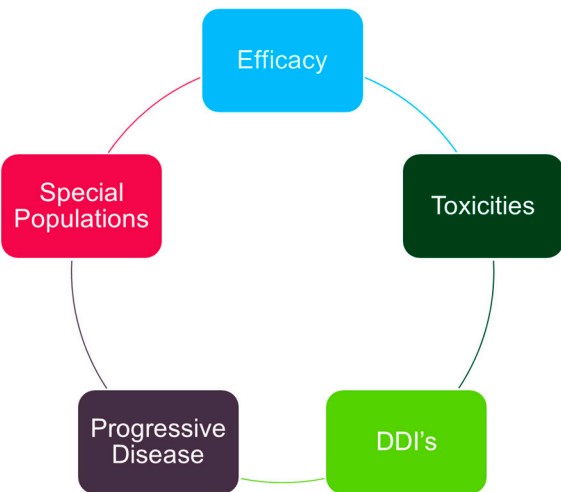

**Figure 5.** Proposed conceptual framework for the management of metastatic prostate cancer. DDI's: drug–drug interactions.

Next, thyroid dysfunction may be observed at a higher frequency in patients receiving apalutamide but is uncommonly clinically significant. In TITAN, 6.5% of participants had all-grade (non-severe) hypothyroidism, autoimmune thyroiditis or blood thyrotropin elevation relative to 1.1% in the placebo group [22]. The authors recommend every 3–6-month serum thyrotropin, free thyroxine and free triiodothyronine in patients receiving apalutamide and optimization of clinically abnormal levels with oral thyroid replacement based on standard targets.

Lastly, apalutamide may be associated with early-onset (within 6 months) skin rash, as described in TITAN [21]. Cutaneous reactions tend to be mild and generally self-limiting and can range from a maculopapular rash to severe toxic skin eruptions. In certain circumstances, temporary withdrawal of the offending agent, initiation of topical corticosteroids and referral to dermatology are necessary. For severe drug eruptions, however, withdrawal of the inciting agent along with initiation of systemic corticosteroids with a slow tapering schedule is recommended. Re-challenge with the same agent should then be avoided.

*B. Managing Drug–Drug Interactions*

Apalutamide, enzalutamide and darolutamide are associated with multiple drug–drug interactions that are highly relevant to clinical practice (the Canadian Urological Association 2023 tool card can be found at https://www.cua.org/sites/default/files/Flipbooks/CPD/ddi2023/index.html). Consider the following examples: apalutamide may decrease serum level of rivaroxaban through induction of CYP3A4, thereby decreasing the efficacy of the anticoagulant and its protective effects against cerebrovascular accidents [65]. Therefore, it would be reasonable to consider switching from rivaroxaban to edoxaban to minimize such risk. Similarly, enzalutamide may decrease serum concentrations of both atorvastatin and simvastatin due to potent induction of CYP3A4. Consequently, higher doses of atorvastatin may be necessary to achieve lipid goals [66]. A thorough review of patients' prescriptions is essential to maximize the efficacy of prostate cancer therapy and minimize associated harms related to concomitant co-morbidities.

*C. Determining Progressive Disease*

When managing men with metastatic prostate cancer, it is imperative to re-engage specialists in genitourinary oncology to confirm disease progression and discuss next steps. Furthermore, the identification of prostate cancers that harbor deleterious mutations in homologous recombination repair is highly relevant given the availability of novel agents that target PARP, as discussed above. The Prostate Cancer Clinical Trials Working

Group-3 (PCWG-3) has specified three key features, of which any two are required to establish progressive disease: (1) symptomatic progression, (2) radiographic progression and (3) biochemical progression [67]. Patients should be referred back to the treating medical oncologist or urologist if intolerable adverse events result from treatment in spite of dose adjustments, re-staging imaging confirms new lesions or meets the standard definition for disease progression [68], two or more consecutive increases in PSA are detected or if cancer-related symptoms arise regardless of PSA level.

*D. Notable Considerations in Older Adults*

The International Society of Geriatric Oncology recommends classifying men older than 75 years with prostate cancer as healthy, vulnerable or frail and subsequently tailoring treatment decisions based on individual health status instead of chronological age [69]. Older cancer survivors may have cognitive deficits which negatively affect quality of life and daily functioning and are therefore important concerns when selecting appropriate drug therapy in this vulnerable group [70]. Central nervous system blockade of the androgen pathway may lead to adverse consequences on cognitive function and emotional wellbeing requiring prompt identification and management. A number of reports have addressed the cognitive and psychological effects of ADT and early generation androgen receptor antagonists [71–74]. In a recent systematic review of 15 randomized clinical trials in nearly 9000 men with metastatic prostate cancer addressing cognition and depression effects of ARAT agents, depression was found to be more commonly assessed than cognition in men receiving ARATs [75]. Self-reported depression measures favored abiraterone over enzalutamide and both abiraterone and enzalutamide over placebo but not necessarily over first-generation antiandrogens. Furthermore, abiraterone may improve short-term emotional functioning relative to enzalutamide. The magnitude and clinical importance of its effects on cognition and depression were unclear. Data evaluating apalutamide and darolutamide were not available [75]. The authors believe that it is reasonable to consider annual cognition and depression assessment using validated screening tools such as the Montreal Cognitive Assessment test [76] and Patient Health Questionnaire-9 [77]. Psychosocial oncology supports are expected to improve and maintain quality of life for older adults meeting the criteria for cognitive impairment or depression in order to optimize adherence and oncologic outcomes for this vulnerable cohort.

## 6. Conclusions

Promising new therapeutic strategies for men with metastatic prostate cancer are improving outcomes and have manageable toxicity profiles. General practitioners are encouraged to develop their knowledge and experience using androgen receptor axis-targeted agents as well as managing relevant toxicities in collaboration with medical oncologists. Recommendations from the Canadian Urologic Association, American Urologic Association and European Society of Medical Oncology are available online and are useful resources. Recently, novel therapies in men with metastatic castrate-resistant prostate cancer combining PARP inhibitors, ARAT agents and ADT have further improved outcomes in cancers that harbor germline or somatic HRR defects and require careful identification. Collaborative efforts that include medical oncologists and general practitioners in oncology are needed to optimize care for men with incurable prostate cancer.

**Author Contributions:** Conceptualization, A.B. and M.V.; investigation, A.B., D.G. and M.V.; writing—original draft preparation, A.B. and M.V.; writing—review and editing, A.B., D.G. and M.V. All authors have read and agreed to the published version of the manuscript.

**Funding:** The authors would like to acknowledge that article processing fees were graciously paid by the Canadian Association of General Practitioners Oncology.

**Data Availability Statement:** The authors will provide additional information on their research. For more information, please contact the corresponding author.

**Conflicts of Interest:** A.B. has received honoraria from Astellas, Bayer, Janssen and Astra Zeneca.

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
