# Peer review of "Contemporary Systemic Therapy Intensification for Prostate Cancer: A Review for General Practitioners in Oncology"

_curroncol, doi:10.3390/curroncol31020078_

Round 1

Reviewer 1 Report

Comments and Suggestions for Authors

Batra and Valdes have written a comprehensive overview of novel treatments for non-curable PC. The work is logically disposed and offers a quick guide of the field. For this, the authors should be complemented.

The manuscript does of course, given its nature, lack in originality but will likely be of interest for many and aid in clinical decision-making.

Minor comments:

The discussion is perhaps too local and adjusted to, specifically, Canadian circumstances. In most countries GPs never prescribe or monitor ARATs. Consider re-writing 341-345 (e.g., include EAU guidelines) to acknowledge the international readers.

If GPs are addressed, the continuation/ discontinuation of ADT when novel therapies are prescribed should be more clearly addressed. I.e. explain ADT remains and why.

Author Response

1. The discussion is perhaps too local and adjusted to, specifically, Canadian circumstances. In most countries GPs never prescribe or monitor ARATs. Consider re-writing 341-345 (e.g., include EAU guidelines) to acknowledge the international readers.

  • We have made revisions to the Introduction and Conclusions sections highlighting the ESMO guidelines to differentiate between North American and international providers vis-a-vis prostate cancer management. 
  • See Lines 28-42 and 499-501.

2. If GPs are addressed, the continuation/ discontinuation of ADT when novel therapies are prescribed should be more clearly addressed. I.e. explain ADT remains and why.

  • Comment added to the CRPC section.
  • See Lines 196-202.

Reviewer 2 Report

Comments and Suggestions for Authors

The ambition of the authors is to provide a guide for general practitioners in oncology. This medical specialisation is specific for Canada and this fact must be acknowledged in the manuscript. 

Major revision is needed as follows (line numuber and comment):

Elsewhere: review statistical evaluation which needs standard reporting (RR, 95%CI)

Check citations: e.g. cit. 15 is missinng and several citations are not in agreement.

Many words are missing what makes writing unclear and scientifically insufficient - marked with ? bellow.

26-27: advance prostate cancer is the broader category than metastatic prostate cancer and includes also patients with lower-grade disease with an increased risk or progression (e.g. N1). "Synchronous" and "metachronous" allocation is relevant for metastatic (M1) disease and not "advanced prostate cancer". Please, correct.

63: that same year...no year is mentioned in previous text..

65-71:  Fig. 1 and 2 algorithms are inappropriate. The role of docetaxel in mHSPC treated with ADT and ARTA remains unclear and docetaxel eligibility is not considered a discrimination factor. Moreover, there is no proven benefit of docetaxel in low volume disease. Local treatment (prostate RT) recommendation is missing.

101: OS was reduced?

113: Abiraterone was validated?

126-127: not conventional CT but conventional imaging which includes CT, MR and bone scan

129: second generation androgen inhibitors

137: Aramis trial: cite publication (Nonmetastatic, Castration-Resistant Prostate Cancer and Survival with Darolutamide | NEJM)

120-149 two important points must be mentioned:

- metastasis free survival improvement (accepted surrogate for OS)

- favourable toxicity profile (though toxicity is reasonably covered in the next chapter)

160-206: Docetaxel should be mentioned on the first place, historically and also discriminating pre and post-docetaxel settings

ABI and ENZA registration studies in post-docetaxel setting are not mentioned.

Radiopharmaceuticals (223Ra and 177Lu) are not mentioned.

165 median?

166 relative risk reduction?

197categorization is -was

208-2020 the whole chapter sounds unclear and requires re-writing 

223 In Talapro-2

228 not reached?

235 Magnitude?

246 Propel?

293 CUA -explain

305 - 316: definition of progression is inconsistent with PCWG-3 recommendation: two  factors are required to determine disease progression. In case recommendation for general practitioners in oncology is differrent, this must be supported by a consensus or guidelines and acknowledged.

317 - 323: Central nervous system...the rest of this chapter is not a consideration in elderly patients and should be separated in a new chapter.

Comments on the Quality of English Language

Check wording, scientific meaning and citations.

Author Response

The ambition of the authors is to provide a guide for general practitioners in oncology. This medical specialisation is specific for Canada and this fact must be acknowledged in the manuscript. 

  • We have acknowledged this fact in both the Introduction and Conclusions sections.
  • See Lines 28-42 and 499-501.

Major revision is needed as follows (line number and comment):

Elsewhere: review statistical evaluation which needs standard reporting (RR, 95%CI)

  • Does not apply to this Special Issue.

Check citations: e.g. cit. 15 is missing and several citations are not in agreement.

  • All citations reviewed.

Many words are missing what makes writing unclear and scientifically insufficient - marked with ? bellow.

26-27: advance prostate cancer is the broader category than metastatic prostate cancer and includes also patients with lower-grade disease with an increased risk or progression (e.g. N1). "Synchronous" and "metachronous" allocation is relevant for metastatic (M1) disease and not "advanced prostate cancer". Please, correct.

  • Corrected; advanced replaced with metastatic to be consistent throughout the manuscript.

63: that same year...no year is mentioned in previous text.

  • Corrected, see Line 66-67

65-71:  Fig. 1 and 2 algorithms are inappropriate. The role of docetaxel in mHSPC treated with ADT and ARTA remains unclear and docetaxel eligibility is not considered a discrimination factor. Moreover, there is no proven benefit of docetaxel in low volume disease. Local treatment (prostate RT) recommendation is missing.

  • Figures 1 and 2 amended to reflect data limitations.
  • Limitations vis-à-vis use of triplet therapy is discussed in the text, see Lines 155-164)
  • Prostate RT for low volume disease discussed in the text, see Lines 168-175).

101: OS was reduced?

  • Corrected, Line 110.

113: Abiraterone was validated?

  • Corrected, Line 145.

126-127: not conventional CT but conventional imaging which includes CT, MR and bone scan

  • Corrected, Lines 184-186.

129: second generation androgen inhibitors

  • Corrected, Line 188.

137: Aramis trial: cite publication (Nonmetastatic, Castration-Resistant Prostate Cancer and Survival with Darolutamide | NEJM)

  • Corrected, Lines 192-196, references 30-31).

120-149 two important points must be mentioned:

- metastasis free survival improvement (accepted surrogate for OS)

- favourable toxicity profile (though toxicity is reasonably covered in the next chapter)

  • - Does not apply to this Special Issue.

160-206: Docetaxel should be mentioned on the first place, historically and also discriminating pre- and post-docetaxel settings

  • Docetaxel data placed prior to ARAT data. Re-organized text to discriminate between pre- and post-docetaxel settings.
  • Lines 233-247.

ABI and ENZA registration studies in post-docetaxel setting are not mentioned.

  • Added to section 4.
  • Lines 287-339.

Radiopharmaceuticals (223Ra and 177Lu) are not mentioned.

  • Data pertaining to radiopharmaceuticals added to section 4D.
  • Lines 392-421.

165 median?

  • Corrected, Line 292

166 relative risk reduction?

  • HR for death of 0.81 or relative risk reduction in death of 19% are equivalent statements.
  • Line 293.

197categorization is -was

  • Line 245.

208-2020 the whole chapter sounds unclear and requires re-writing 

  • Does not apply to this Special Issue.

223 In Talapro-2

  • Sentence clarified.
  • Line 358

228 not reached?

  • Does not apply to this Special Issue.

235 Magnitude?

  • Sentence clarified.
  • Line 370.

246 Propel?

  • Sentence clarified.
  • Line 382.

293 CUA -explain

  • CUA comment explained.
  • Lines 464-465

305 - 316: definition of progression is inconsistent with PCWG-3 recommendation: two factors are required to determine disease progression. In case recommendation for general practitioners in oncology is differrent, this must be supported by a consensus or guidelines and acknowledged.

  • Comment added pertaining to PCWG-3 recommendations specifying two factors being necessary for determination of progressive disease.
  • Lines 480-483.

317 - 323: Central nervous system...the rest of this chapter is not a consideration in elderly patients and should be separated in a new chapter.

  • Does not apply to this Special Issue.

Reviewer 3 Report

Comments and Suggestions for Authors

Dear Author

This study was reported the management of metastatic prostate cancer. The reviewer agrees to some of the content. However, the reviewer would like to suggest some opinions to make this paper as follows.

Major point

1.     The author should change the strategy for mCSPC. The superiority for triplet therapy consisted of docetaxel and darolutamide are controversial. The comparative study with dubblet therapy consisted of ARSI and triplet therapy is none. In figure 1 and 2, triplet therapy is expressed all comer except docetaxel unfit or decline.

2.     The author should refer the radiotherapy for low volume mCSPC and MDT.

3.     The author should indicate LATITUDE trial.

4.     The author should show ARASENSE trial.

5.     Treatment strategies for nmCRPC need to be reconsidered, such as the case of the biochemical recurrence after radical prostatectomy/radiotherapy or with lymph node positive.

6.     Treatment strategies for mCRPC need to be reconsidered, such as the timing of genomic testing.

7.     The author should refer cabazitaxel.

Minor point

Page 1 line 27 “recur” may be “recurrense”.

Author Response

This study was reported the management of metastatic prostate cancer. The reviewer agrees to some of the content. However, the reviewer would like to suggest some opinions to make this paper as follows.

Major point

  1. The author should change the strategy for mCSPC. The superiority for triplet therapy consisted of docetaxel and darolutamide are controversial. The comparative study with doublet therapy consisted of ARSI and triplet therapy is none. In figure 1 and 2, triplet therapy is expressed all comer except docetaxel unfit or decline.
  • Figures 1 and 2 amended to reflect data limitations.
  • Limitations vis-à-vis use of triplet therapy is discussed in the text, see Lines 155-164)
  1. The author should refer the radiotherapy for low volume mCSPC and MDT.
  • Prostate RT for low volume disease discussed in the text, see Lines 168-175).
  1. The author should indicate LATITUDE trial.
  • Corrected, reference #7.
  • Lines 85-92.
  1. The author should show ARASENSE trial.
  • Added, see reference #25
  • Lines 135-142.
  1. Treatment strategies for nmCRPC need to be reconsidered, such as the case of the biochemical recurrence after radical prostatectomy/radiotherapy or with lymph node positive.
  2. Does not apply to this Special Issue.
  3. Treatment strategies for mCRPC need to be reconsidered, such as the timing of genomic testing.
  4. Does not apply to this Special Issue.
  5. The author should refer cabazitaxel.
  • Added, Lines 259-286.

Minor point

Page 1 line 27 “recur” may be “recurrence”.

  1. We prefer recur.

Round 2

Reviewer 1 Report

Comments and Suggestions for Authors

Revisions are sufficient.

Author Response

Revisions are noted to be sufficient.

Reviewer 2 Report

Comments and Suggestions for Authors

The manuscript has been completed according to the revision recommendations and provides more comprehensive and consistent information now, so it can be accepted for publication. 

Comments on the Quality of English Language

Please check minor English mistakes and a few missing words.

Author Response

All English errors and missing words have been addressed.

Reviewer 3 Report

Comments and Suggestions for Authors

This study was reported systemic therapy for metastatic prostate cancer. It was a good overview of the evolution of prostate cancer treatment. However, the reviewer would like to suggest some opinions to make this paper as follows.

Major point

1.     Perhaps the title “Systemic Therapy for Metastatic Prostate Cancer: A Review for General Practitioners in Oncology” should be changed because this paper contains of “Non-metastatic castrate resistant prostate cancer” section.

2.     The author should show PROfound trial in Novel therapeutic strategies targeting poly ADP-ribose polymerase section.

Minor point

1.     In Figure 1, is the asterisk in the correct position? The reviewer considers “Docetaxel-fit*” is right or the solid line should change dotted line to “Docetaxel-fit”.

2.     “Radiotherapy for low volume disease” section is not included in “Docetaxel with oral hormonal agents and ADT (“triplet-therapy”)” section.

Author Response

  1.  Perhaps the title “Systemic Therapy for Metastatic Prostate Cancer: A Review for General Practitioners in Oncology” should be changed because this paper contains of “Non-metastatic castrate resistant prostate cancer” section.
    1. Agree, new title provided.
  2.  The author should show PROfound trial in Novel therapeutic strategies targeting poly ADP-ribose polymerase section.     
    1. PROfond data added to Section 4.C.1.
  3.  In Figure 1, is the asterisk in the correct position? The reviewer considers “Docetaxel-fit*” is right or the solid line should change dotted line to “Docetaxel-fit”.
    1. Position of the asterisk has been corrected.
  4.  “Radiotherapy for low volume disease” section is not included in “Docetaxel with oral hormonal agents and ADT (“triplet-therapy”)” section.
    1. A separate sub-section has been added for these data, see Section 2.D.

Round 3

Reviewer 3 Report

Comments and Suggestions for Authors

This study was reported systemic therapy for prostate cancer. The report is easy to understand.